# Uncertainty in non-CO$_2$ greenhouse gas mitigation contributes to ambiguity in global climate policy feasibility

Mathijs Harmsen [1,2] ✉, Charlotte Tabak [1], Lena Höglund-Isaksson [3], Florian Humpenöder [4], Pallav Purohit [3] & Detlef van Vuuren [1,2]

Despite its projected crucial role in stringent, future global climate policy, non-CO$_2$ greenhouse gas (NCGG) mitigation remains a large uncertain factor in climate research. A revision of the estimated mitigation potential has implications for the feasibility of global climate policy to reach the Paris Agreement climate goals. Here, we provide a systematic bottom-up estimate of the total uncertainty in NCGG mitigation, by developing 'optimistic', 'default' and 'pessimistic' long-term NCGG marginal abatement cost (MAC) curves, based on a comprehensive literature review of mitigation options. The global 1.5-degree climate target is found to be out of reach under pessimistic MAC assumptions, as is the 2-degree target under high emission assumptions. In a 2-degree scenario, MAC uncertainty translates into a large projected range in relative NCGG reduction (40–58%), carbon budget (±120 Gt CO$_2$) and policy costs (±16%). Partly, the MAC uncertainty signifies a gap that could be bridged by human efforts, but largely it indicates uncertainty in technical limitations.

Roughly one-third of present-day global warming can be attributed to non-CO$_2$ greenhouse gases (NCGGs), such as *methane* (CH$_4$), *nitrous oxide* (N$_2$O) and *fluorinated greenhouse gases* (HFCs, PFCs, SF$_6$ and NF$_3$)[1]. Correspondingly, reaching ambitious climate targets also requires deep reductions of these gases[2,3]. Reducing NCGG emissions as part of a mitigation strategy can have substantial benefits, including (1) cost reductions[4–14], (2) rapid impacts on temperature (given the short lifetimes of some NCGGs[5], and (3) substantial health benefits, as several gases are also air pollutants[15]. Nevertheless, most attention in climate policy analysis has been paid to CO$_2$, given its large share in overall emissions[16].

Global climate change mitigation research relies heavily on integrated assessment models (IAMs)[17]. For projected NCGG mitigation, these IAM models almost universally use NCGG marginal abatement cost (MAC) curves. These are region- and source-specific datasets used in climate policy research and scenario development to estimate emission reduction potentials and costs. Comprehensive sets of long-

term MAC curves are rarely produced, and many models use relatively old information[18,19]. (See Supplementary S1 for an overview of the MAC data used for a selection of IAMs). Moreover, IAMs typically use only 'one' middle-of-the-road estimate. Therefore, the inherently high uncertainty and possible large consequences for climate policy are largely unknown or at least hidden in most climate change mitigation scenarios.

This study aims to understand the uncertainty in the mitigation potential of emissions from all major NCGG emission sources and the implications for climate policy feasibility, strategies and costs. For this, we develop 'optimistic', 'pessimistic', and default NCGG MAC curves based on a comprehensive literature review, representing the uncertainty range in relative emissions reductions. We subsequently assess the implications of the MAC curve uncertainty in meeting the objectives of the Paris Agreement using the IMAGE 3.2 integrated assessment model[20,21] (Supplementary S2). By varying assumptions on human activities, this setup also allows an assessment of the impact of

[1]PBL Netherlands Environmental Assessment Agency, Bezuidenhoutseweg 30, NL-2594 AV The Hague, the Netherlands. [2]Copernicus Institute of Sustainable Development, Utrecht University, Princetonlaan 8a, NL-3584 CB Utrecht, the Netherlands. [3]Pollution Management Group, International Institute for Applied Systems Analysis, A-2361 Laxenburg, Austria. [4]Potsdam Institute for Climate Impact Research (PIK), Member of the Leibniz Association, Potsdam, POBox 60 12 03, D-14412 Potsdam, Germany. ✉e-mail: mathijs.harmsen@pbl.nl

human activities on overall uncertainty, next to the implications from technical uncertainty represented by the MACs.

The MACs represent all major emitting sectors: agriculture, industry, waste, and fossil fuel production. (See methods and Supplementary S3). They have been developed using the method by ref. 9 but complemented with uncertainty ranges and the inclusion of an additional approx. 120 studies on mitigation measures. The MAC uncertainty analysis is performed with the most detail for the agricultural sources since (1) these are hardest to abate (and thus most relevant in stringent climate scenarios)[18], (2) mitigation potentials are most uncertain, and (3) can be based on the fully bottom-up approach by ref. 9, with quantitative estimates for all underlying parameters. The agricultural MACs are built-up from quantitative components, representing (1) reductions when measures can be applied, (2) technical applicability, (3) non-technical implementation barriers, (4) technological progress, (5) correction for overlap between measures and (6) costs (See Methods and Supplementary S4). For each component, uncertainty ranges have been estimated, where possible, based on literature from up to and including 2022. In a Monte Carlo (MC) simulation, these input parameters have been varied to determine the lower and upper bounds of the overall relative reduction potential per emissions source. For all non-agricultural sources, uncertainty has been estimated by deriving source-specific maximum reduction potentials from literature and expert insights from the GAINS research group[22,23] (see Methods and Supplementary S5). A full MC analysis is not possible for these sources, since most values of the underlying parameters are unknown, as the short-term MAC data is based on external databases[23-25]. However, reduction potentials for non-agriculture sources are generally higher than for agriculture sources (measures are typically more applicable for targeting source emissions, with higher reductions when applied), implying lower uncertainty and resulting in lower residual emissions in stringent climate scenarios[9,18]. All MAC curves are available for further research (including model-based analysis). See Supplementary Data File 1.

# Results
## Agricultural measures
The main goal of the literature study has been to include recent case studies on agricultural measures to the former dataset[9] by collecting information on reduction efficiencies (RE), technical applicability (TA) and costs. RE represents the relative emission reduction when a measure is applied. TA represents the share of the baseline emissions where a measure can be applied. Table 1 gives an overview of the included measures and associated RE values (Supplementary S6 includes a table with all emission sources and a description of the measures and assumptions for all emission sources). Several agricultural sources included in ref. 9 have been excluded here because they are implicitly part of other measures or conflict with them ($CH_4$ enteric fermentation: Improved milk production, extended productive life and for $N_2O$ fertilizer: fertilizer free zone, sub-optimal fertilizer application). The following additional measures have been included in this study: for $CH_4$ enteric fermentation: Seaweed asparagopsis taxiformis as a feed supplement (optimistic case only); for $CH_4$ manure: solid-liquid separation; for $N_2O$ fertilizer: Biochar (optimistic case only), no-tillage, irrigation practices, and for $N_2O$ manure: Anaerobic digestion and manure acidification.

Next to collecting data on RE values (Table 1), the literature study also contributed to updating the default assumptions for the components TA[26,27] and costs[28-40]. Supplementary S7 provides an overview of all input values to the Monte Carlo analysis.

## Optimistic/default/pessimistic MAC curves
The 'optimistic', default and 'pessimistic' MAC curves have been developed for all major NCGG sources for 26 world regions and the 2020–2100 period (See Supplementary Data File 1. Figure 1 shows

the MAC curves for the five agricultural sources (for example: Western Europe). See Supplementary S8 for an overview of the non-agricultural MACs ($CH_4$ and $N_2O$). As the approach and part of the data were similar to those used in ref. 9, it is relevant to compare the maximum reduction potentials (MRPs) of the MACs in both studies (see also Supplementary S9 with an MRP comparison for all sources in 2050 and 2100). For the agricultural sources, the ref. 9 default estimate is generally found between this study's default and optimistic value, i.e., this study's default reduction potential is generally somewhat lower. $N_2O$ emissions from manure form an exception with a slightly higher MRP due to newly included measures. This is mainly the result of the Monte Carlo approach used in this study, where lower implementation and technical applicability values are included in the solution space. For $CH_4$ rice, recent studies[41,42] also indicate a lower reduction efficiency. Further, this study assumes a higher overlap between $CH_4$ manure measures.

## Scenario analysis
The MAC curves have been used as an input to IMAGE in conjunction with Shared Socio-economic Pathway (SSP) based scenario assumptions[43]. The scenarios are described in Table 2. The core set to assess the implications of the MAC uncertainty is based on SSP2, a scenario with middle-of-the-road socio-economic and technological development assumptions. The scenarios are set to reach a 1.5- and 2-degrees Celsius target in 2100 (represented by 2.0 W/m² and 2.6 W/m² radiative forcing targets) under optimistic, default and pessimistic NCGG MAC assumptions (i.e., with high (H), medium (M) and low (L) reduction potentials, respectively). The mitigation scenario implications are compared to a no climate policy baseline (Base). Pre-2100 temperature overshoots are allowed. The SSP2-based 2-degree scenarios follow the nationally determined contributions (NDCs) until 2030, followed by fragmented regional climate policy until 2040 and globally concerted climate action until 2100 (i.e., category C3b in the IPCC's scenario classification[44]). The 1.5-degree scenarios are of category C2 (allowing a temperature overshoot). These scenarios also allow for increased pre-2030/2040 climate ambition additional to the NDCs.

In addition, the analysis includes two additional SSP scenarios (in a 2-degree case) to assess the additional uncertainty due to human activities: SSP1 and SSP3, with low and high GHG-emitting activities, respectively (see methods for underlying scenario assumptions). SSP1 is combined with optimistic MAC assumptions (H) and SSP3 with pessimistic assumptions (L) to represent the extremes in NCGG emissions. The goal of the scenario analysis is to analyze the effect of MAC uncertainty and uncertainty in human NCGG emitting activities on:

- Feasibility of scenarios
- NCGG emission reductions (total and source-specific)
- Climate policy costs
- Remaining global carbon budgets, i.e., the need for $CO_2$ mitigation

The scenarios used to assess uncertainty in GHG-emitting activities (2H_SSP1 and 2L_SSP3) have only been used for the feasibility and carbon budget calculations. Policy costs and NCGG reduction are not directly comparable due to different cost and baseline emission assumptions.

## Climate targets are out of reach under pessimistic assumptions
Of the scenarios described in Table 2, both 1.5 L and 2L_SSP3 have proven to be infeasible, when using the IMAGE model setup. This implies that under pessimistic NCGG mitigation assumptions, the 1.5-degree climate target cannot be reached, despite maximum climate policy efforts. Further, the combination of high GHG-emitting activities (SSP3-based) and a low NCGG mitigation potential would even keep the 2-degree climate target out of reach. Note that these

**Table 1 | Included agricultural reduction measures, associated reduction efficiencies (when fully applied) and underlying literature**

| | Measures | Range in reduction efficiencies (%) | References |
|---|---|---|---|
| $CH_4$ - Enteric fermentation | Addition of nitrate to the feed | 21–42 | 58–65 |
| | Genetic selection and breeding | 8–31 | 66–70 |
| | Adding tannins as a food supplement | 10–32 | 71–75 |
| | Grain processing | 10–38 | 73,76–78 |
| | Improved health monitoring and illness prevention | 4–20 | 28,68,79,80 |
| | Seaweed (Asparagopsis taxiformis) | 12–99.5 | 81–88 |
| $CH_4$ - Rice production | Rice straw mitigation | 26.5–61 | 29–31,89–91 |
| | Direct seeding | 16.6–47 | 29,91–94 |
| | Replacing urea with ammonium sulfate | 14.18–42 | 29,91,95,96 |
| | Addition of phosphogypsum | 28–86 | 29,91,97–100 |
| | Alternate flooding and drainage | 18.8–79 | 29,31,32,41,42,74,91,101–115 |
| $CH_4$ - Manure | Manure acidification | 61–98 | 73,90,116–120 |
| | Anaerobic digestion | 25–75 | 29,121–123 |
| | Solid-liquid separation | 46–81 | 121,122 |
| | Manure storage: duration | 38–76 | 124 |
| | Housing systems and beddings | 4–96 | 58,73,125–129 |
| | Manure storage covering | 0–90 | 58,73,118,130 |
| $N_2O$ - Fertilizer | Nitrification inhibitors | 17–60 | 53,58,131–141 |
| | Improved land manure application | 5–50 | 33,138,142–145 |
| | Irrigation practices | 15–67 | 146–149 |
| | Biochar | 14–38 | 150–153 |
| | Spreader maintenance | 22–42 | 13,29,154,155 |
| | Improved agronomy practices | 14–54 | 33,156–161 |
| | No-tillage | 25–48 | 162–166 |
| $N_2O$- Manure | Reduced dietary protein | 0–52 | 73,167–171 |
| | Decreased manure storage time | 35–35 | 73 |
| | Manure storage covering | 30–75 | 58,73 |
| | Improved animal housing systems and bedding | 9–88 | 58,125,127,128 |
| | Anaerobic digestion | 34–75 | 123,172,173 |
| | Acidification | 0–96 | 174–179 |

conclusions depend on the use of the model. Model comparisons have shown that compared to other models, IMAGE can be regarded as average in terms of inertia/speed of implementation and energy system transformation[45]. This automatically implies that some models may still find the 1.5-degree target within reach based on more optimistic assumptions. Note further that, given that the world is close to exceeding 1.5 degrees warming, there are multiple factors that can be considered 'make-or-break' for reaching the 1.5-degree target, such as the level of near-term $CO_2$ reduction and $CO_2$ removal.

Figure 2 shows the results from the scenario exercise. Optimistic NCGG assumptions (indicated in light green) correspond with high NCGG reductions, lower policy costs and higher carbon budgets, with opposite relations under pessimistic assumptions (indicated in orange).

**Range in NCGG reduction**

Unsurprisingly, MAC uncertainly results in considerable ranges in projected NCGG reductions (panel a) (see also Supplementary S10 for the emission trajectories). This is indicated by the range under the same (SSP2) baseline assumptions, with (in relative difference with a no climate policy baseline in $CO_2$ equivalents, in 2100) 40% to 58% in the 2-degree case and 53–65% in the 1.5-degree case. Net NCGG reductions only provide an overall indication because of the policy-dependent choice of GWP metric (here: AR4 $GPW_{100}$) to convert NCGG emissions to $CO_2$ equivalents. Supplementary S11 gives the source-

specific relative and absolute reductions. *Methane* mitigation is the main contributor to total NCGG reduction (in 2100: 45–51%), followed by HFCs (31–38%), $N_2O$ (13–17%) and small contributions of $SF_6$ (1.7%) and PFCs (0.5%). In all mitigation scenarios, total F-gases are reduced by more than 90% in 2100, leaving most of the uncertainty with $CH_4$ and $N_2O$. The gas-specific uncertainty is also reflected by differences in the climatic influence of individual gases. The projected (MAGICC6.3-based) difference in high vs. low radiative forcing in a 2-degree case in 2100 is for (in W/m2): $CH_4$: 0.08, $N_2O$: 0.05, F-gases: 0.02. In other words, even with pessimistic F-gas assumptions, residual emissions are expected to be low. Uncertainty is relatively high for PFCs and $SF_6$ compared to HFCs, but their net effect is small due to their relatively low share in total emissions. An average 57% of total $CH_4$ reductions is realized in fossil energy. However, the scenario differences are largely defined by differences in projected agriculture emissions. This is also the case for $N_2O$ where 90% of the emissions are produced in agriculture.

Scenario differences in emission reductions increase over the century as the average and range in mitigation potentials in the MACs increase. We find no significant impact of MAC uncertainty on peak warming, due to the early-century similarities between the emission trajectories. The maximum radiative forcing levels (typically peaking between 2030 and 2040) and maximum global mean temperatures are very similar across the (SSP2-based) 2-degree scenarios and across the 1.5-scenarios (see Supplementary S10). Note however, that peak

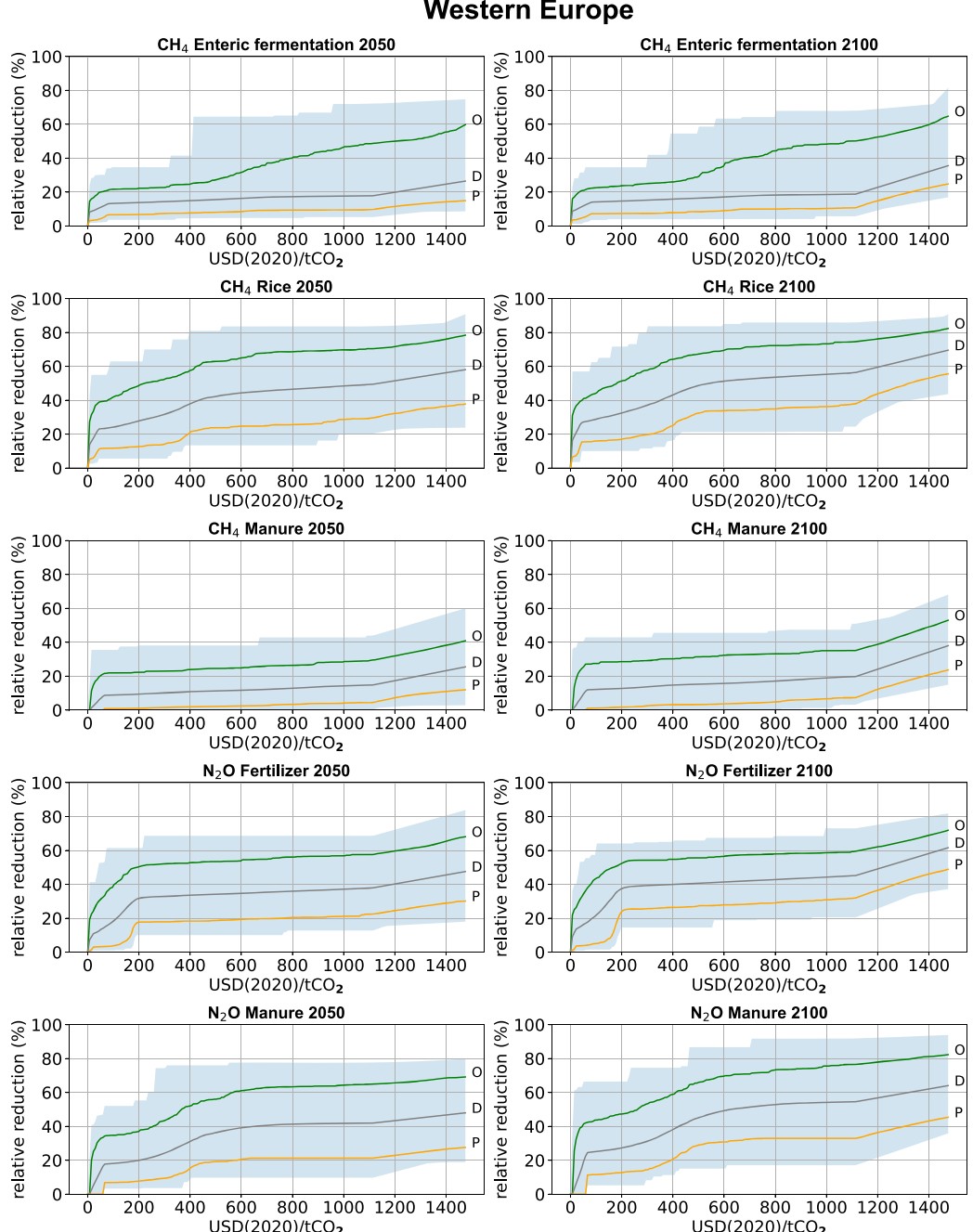

**Fig. 1 | Agricultural MAC curves. Example: Western Europe.** Optimistic (green), default (gray) and pessimistic (orange) MACs represent the 5th, 50th, and 95% percentile in a 1000 MAC range. The blue-shaded area shows the Monte Carlo range. Left panels: 2050, Right panels: 2100. Relative reduction (*Y*-axis) is relative to the present-day, global mean emission intensity. $CO_2$ eq. prices (*X*-axis) are given in 2020$.

temperature is found to be slightly (0.02 degrees) lower in the 2H_SSP1 case, due to earlier allowed action (ratcheting up the NDCs) and lower (SSP1) baseline emissions. The impact of NCGG mitigation potential on peak temperature is model- and scenario dependent and could be further explored in a multi-model study.

## Climate policy costs
Global climate policy costs (Fig. 2b) strongly depend on the availability of NCGG mitigation options, which are on average lower in cost than $CO_2$ mitigation options[9], but also expand the range of possible measures. When low-cost options are exhausted earlier (i.e., in the pessimistic MAC case), climate targets can only be met by applying higher-cost mitigation measures (both for $CO_2$ and NCGG emissions). This is

indicated by the 32% difference in cost between the pessimistic and optimistic 2-degree scenarios and a 42% difference between the default and optimistic 1.5-degree scenarios, where nearly all options need to be applied. Although the absolute policy costs are highly uncertain (here, estimated at roughly 1–2% of global GDP), the relative scenario differences give a more robust indication of the large implications of NCGG MAC uncertainty.

## Carbon budgets
Under equal climate targets, cumulative $CO_2$ emissions need to compensate for differences in NCGG emissions, which can be expressed in an allowable global $CO_2$ budget for the remainder of the century (Fig. 2c). The carbon budgets of the 1.5-degree and 2-degree scenarios

**Table 2 | Scenario setup**

| Scenario | NCGG MAC reduction potential | Human GHG-emitting activities | Radiative forcing target 2100 (W/m²) |
|---|---|---|---|
| Base | n.a. | Medium (SSP2) | n.a. |
| 2H | High/Optimistic | Medium (SSP2) | 2.6 |
| 2M | Medium | Medium (SSP2) | 2.6 |
| 2L | Low/Pessimistic | Medium (SSP2) | 2.6 |
| 1.5H | High/ optimistic | Medium (SSP2) | 2.0 |
| 1.5M | Medium | Medium (SSP2) | 2.0 |
| 1.5L | Low/Pessimistic | Medium (SSP2) | 2.0 |
| 2H_SSP1 | High/Optimistic | Low (SSP1) | 2.6 |
| 2L_SSP3 | Low/Pessimistic | High (SSP3) | 2.6 |

Scenarios are SSP2 based, unless otherwise specified under Scenario. No target is set for Base. The IMAGE SSP2 baseline results in a forcing level of 6.2 W/m² in 2100. 1.5 L and 2L_SSP3 are infeasible scenarios (further discussed in Results).

fit within the cumulative $CO_2$ range of the AR6's scenario classification[44], (C2 (1.5-degree with overshoot) –90 to 620 Gt, C3b (NDCs and 2-degree) 560–1050). This study's 1.5-degree scenarios are developed with the aim of having >66.6% chance of staying below 1.5 degrees, whereas the C2 category also allows 1.5-degree scenarios that have a > 50% chance of staying below 1.5-degrees. This also explains that the carbon budget of 76 Gt in 1.5M is on the low side of the range.

MAC uncertainty alone translates into a 240 Gt $CO_2$ range in the carbon budget under 2-degree conditions. Lower (SSP1-based) GHG-emitting activities can increase this value by a projected 38 Gt. No feasible low-enough carbon budget (i.e., level of $CO_2$ mitigation) can be found under the high-emitting, low mitigation conditions in 2L_SSP3. MAC uncertainty is projected to result in a (partial) 184 Gt range in the carbon budget in the 1.5-degree case. The carbon budget estimates from this study's bottom-up uncertainty analysis are relatively consistent with top-down analyses of large scenario ensembles. As part of the IPCC's 1.5-degree Special Report and more recent 6th Assessment Report, it has been estimated that uncertainty in future NCGG emissions could affect the global carbon budget by ±250 Gt $CO_2$ or ±220 Gt $CO_2$, respectively[44,46]. Here, we find a slightly smaller range in a 2-degree case only and with a single model. The large disadvantage of the top-down approach is the difficulty in distinguishing between factors underlying the range. These could also simply be the exclusion of emission categories in models or a simplified representation of NCGG emissions, next to assumptions on activities and mitigation options. Regardless, both the top-down and bottom-up estimates portray NCGGs as a huge uncertain factor, considering the remaining $CO_2$ budgets of roughly 1000 Gt and 400 Gt in a 2-degree and 1.5-degree case, respectively.

## Discussion

This study shows the crucial role that NCGG mitigation needs to fulfill in future stringent climate change mitigation scenarios. It also makes clear that uncertainty in future NCGG mitigation implies that we cannot be confident about the feasibility of stringent climate goals. More NCGG mitigation measure deployment, case studies and research can help in three ways in this respect: (1) It maximizes learning and thus reduction potentials, while lowering costs (2) It stimulates early action, limiting short-term climate change and avoiding limitations in longer-term upscaling, and (3) It helps understand the limitations of NCGG mitigation, leading to more accurate and effective policy strategies.

The MAC curves exclude natural emission sources that can be influenced by human influence, most importantly, $CH_4$ from wetlands. The human-induced GHG emission fluxes (notably from $CH_4$ and $CO_2$) from wetlands are highly uncertain and could either be net positive or negative[47]. This study also excludes uncertainties in NCGG

atmospheric chemistry and climate effects. For all non-included factors, we assumed default values, implying that the uncertainty range is larger in both positive and negative directions, making it likely that NCGG uncertainty has even larger implications for climate policy feasibility.

There are critical differences between the NCGG MAC curves in this study and those developed by US-EPA[24] and GAINS[10,13,14]. The latter MAC datasets mainly represent the present-day technical reduction potentials as measured in multiple case studies, although they do account for modest technical progress towards 2050 (the studies' end year), yet not for changes in the level of technology acceptance. In this study, we deliberately fully account for all future technological change and removal of non-technical implementation barriers under stringent climate policy conditions. The longer-term (up to 2100) perspective of this study also requires that these factors are included, including the high uncertainty that comes with them. As these factors contribute to more effective mitigation, this study's default MAC curves generally represent higher reduction potentials, while this study's pessimistic MACs are generally found to be in line with US-EPA and GAINS (when looking at 2050). This fits well with the assumption that present day reduction potentials should at the very least be reachable in any future scenario, as with the prerequisite that this study's MAC range should span the full potential solution space.

Note that the MAC curves solely specify relative reductions at different price levels. They are agnostic about the likelihood of climate ambitions, which are almost certainly regionally constrained (e.g., lack of finance or ceilings on food prices), represented by the carbon price. These constraints can be estimated exogenously or specified in IAM-based scenario studies. The information in the MACs only represents climate policy implications. Mitigation measures might not be desirable when including non-climate socio-economic aspects (e.g., NCGG pricing leading to higher food prices or negative environmental implications of intensive agriculture).

The MAC curves should only be used as an uncertainty benchmark and explicitly not as a representation of high, default and low ambition levels. It would be misleading to present the optimistic or pessimistic MACs as realistic options that depend on policy choices. To a large degree, the MAC mitigation uncertainty indicates uncertainty in technical limitations, which cannot be influenced by human efforts, whereas the 'human ambition element' should be represented by the carbon price or differences in human activities (represented by different SSP pathways). However, it can be argued that highly uncertain, 'soft' MAC components such as the implementation potential (representing the level of social barriers) or R&D efforts behind technological progress could allow for some minor additional gain at high ambition levels.

## Methods

The method section is structured in four parts: (1) A description of the system boundaries and the coverage of global NCGG emissions, (2) An approach to construct the MACs (provided in more detail in Supplementary S4), (3) The development of the 'optimistic', 'default' and 'pessimistic' MACs (these MAC curves are made available as Supplementary Data 1) and (4) A description of the scenario analysis.

### System boundaries

The MAC curves and scenario assessment in this study are based on the emission source categories of the IMAGE 3.2 model[20,21], representing all anthropogenic NCGGs. IMAGE is an ecological-environmental integrated assessment model (IAM) framework that simulates the environmental consequences of human activities worldwide. It is a partial equilibrium (with price elastic energy and resource demand), simulation model (without foresight). However, a simplified emulator of the model (called FAIR) can be run prior to running the framework to obtain least-cost climate policy data for mitigation scenarios (with a so-called

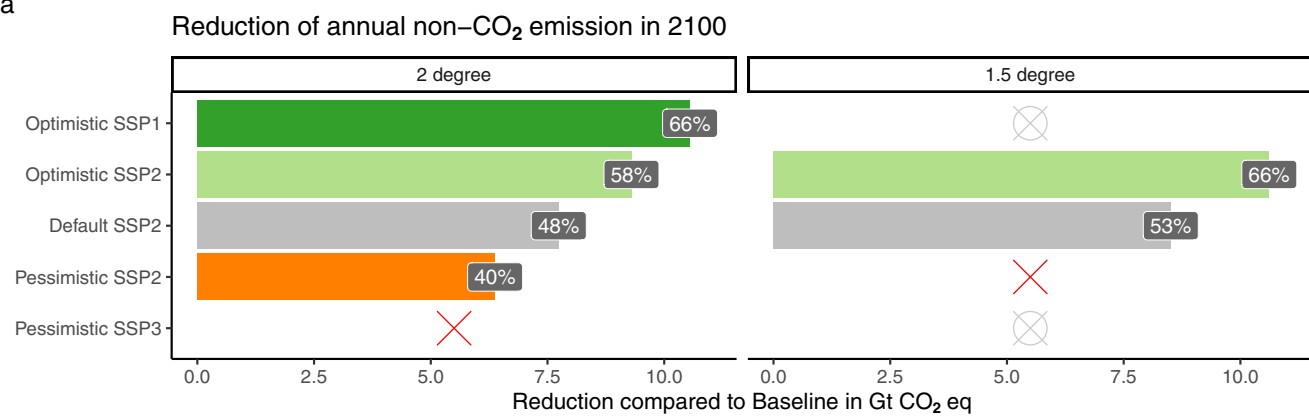

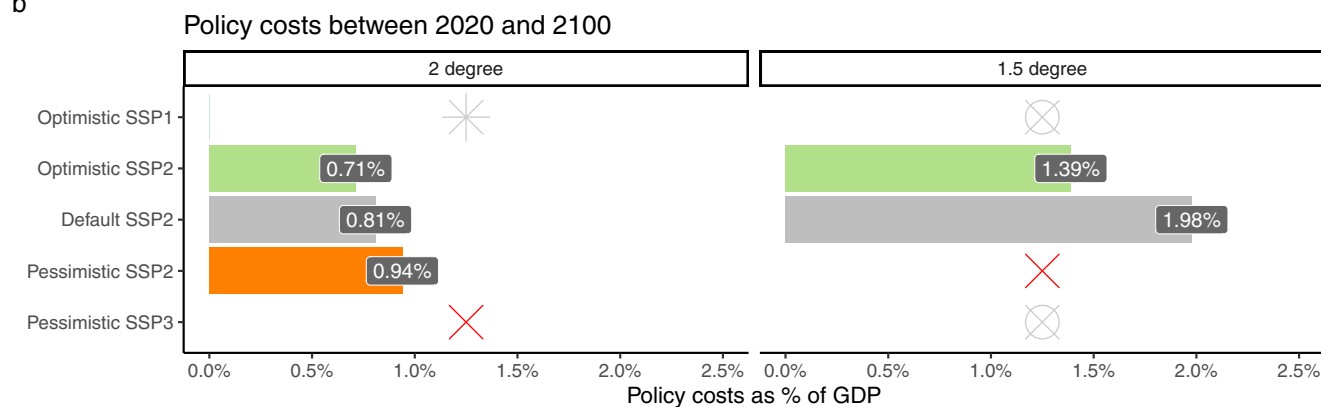

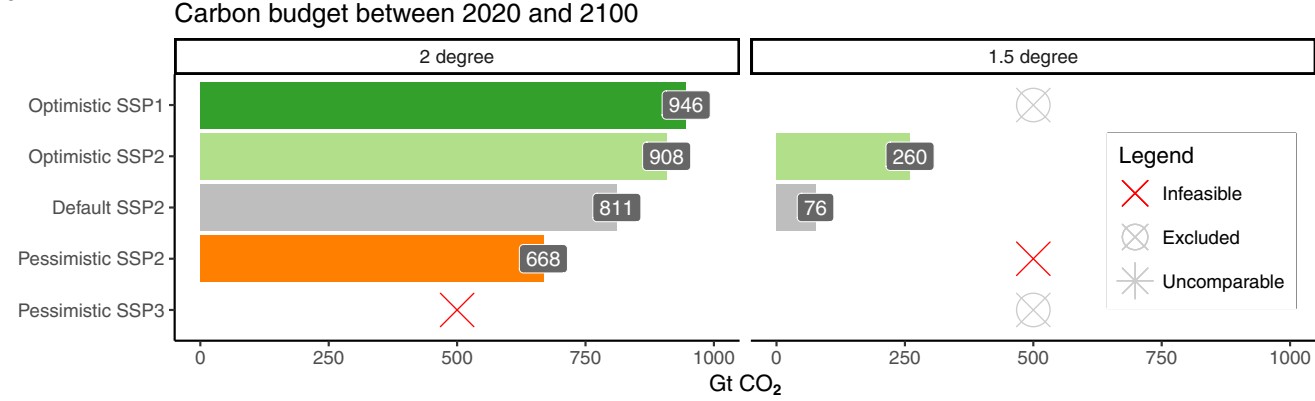

**Fig. 2 | Scenario results.** 2 Degree scenarios: left panels, 1.5-degree scenarios: right panels. NCGG reduction (**a**) shows reduced Gt CO$_2$ equivalents (based on AR4 100-yr GWP) relative to baseline (SSP2) with % reductions in bars. Policy costs (**b**) represent global, first-order direct expenditures as a percentage of global GDP (PPP), discounted over the 2020–2100 period. Discount rate follows the yearly economic growth, with a Ramsey/Stern function. Carbon budgets (**c**) represent the net global CO$_2$ emissions over the 2020–2100 period. Bar colors indicate scenario types: optimistic MAC and low emissions (green), optimistic MAC (light green), default MAC (gray), pessimistic MAC (orange).

recursive-dynamic solution algorithm). IMAGE represents socio-economic developments in 26 world regions to capture spatial and multi-scale differences. IMAGE has a relatively high-detailed land use representation compared to other IAMs. Land use, land cover, and associated biophysical processes are treated at a (5 × 5 arcminutes = 10 × 10 km at the equator) grid level to capture local dynamics. IMAGE uses the reduced-complexity climate model emulator MAGICC6[48] to develop climate change mitigation pathways aimed at reaching climate targets. Calculated climate policy costs in mitigation scenarios represent first-order expenditures (i.e., the 'area under the MAC curve') and exclude further economic impacts on the global economy. See Supplementary S2 for further model information.

The MAC curves in this study cover 92% of the present-day NCGG emissions and 96% of the projected emissions in 2100 (see Supplementary S3). The MAC curves represent potential emission reductions under CO$_2$ equivalent (eq.) prices up to 4000 \$(2005)/tCeq. (or 1446 \$(2020)/tCO$_2$eq.), the maximum price that is applied in the IMAGE IAM framework. Emissions and emission reductions are calculated for the 26 global IMAGE regions. Regional differences in present-day emission intensities and activities are fully represented in the scenario assessment. Regional emissions in the base year (2015 to 2020, depending on the source) are calibrated with data from several detailed databases covering different emissions sources; CEDS[49], GAINS[23], EDGAR 4.2.3[50,51].

## Construction of the MAC curves

The MACs are built up from individual source-specific measures and assumptions on long-term developments (See Supplementary S4 for a more detailed description). The relative reduction potential (RP) (in %) of each mitigation measure in year $t$ and region $r$ is determined by Eq. 1. The maximum reduction potential (MRP) (in %) is the maximum relative abatement compared to baseline source emissions when all source-specific measures are implemented (Eq. 2).

$$RP_{(t,r)} = RE^* TA_{(r)}{}^* OVcorr_{(t,r)}{}^* IP_{(t)} \qquad (1)$$

$$MRP_{(t,r)} = (RP_{1(t,r)} + RP_{2(t,r)} + RP_{3(t,r)} \ldots + RPx_{(t,r)})^* TP_{(t)} - Bcorr_{(t,r)} \qquad (2)$$

With (all in %): TA: Technical applicability, this is the part of the baseline that can technically be covered by the measure. This is often 100%, but can be lower, e.g., if only a sub-process is targeted or if regional climatic circumstances are partly unsuitable. RE: Reduction efficiency, i.e., the relative reduction in case a measure can be applied, generally based on multiple case studies. IP: Implementation potential represents (the lack of) non-technical barriers. This is assumed to increase in time due to improved technology diffusion and policy acceptance. OVcorr: Correction for overlap between measures that target the same emissions. If a subsequent measure is applied, it has a diminished benefit due to lower remaining emissions. Note that this correction increases with time as IP increases (based on ref. 52, see Supplementary S2). TP: Technological progress, increase of the reduction potential with time as a result of new or improved technologies. This is the only factor that is larger than 100% (see Supplementary S2). Bcorr: Correction for regional emission reductions that already occur in the baseline scenario, e.g., due to zero or negative cost measures, such as the use of fugitive $CH_4$ emissions as an energy source, or non-climate policy reductions, such as from air quality measures.

The combination of measures with the highest estimated maximum reduction potential is used to construct MAC curves. It is assumed that the least costly measures are implemented first. When multiple measures are used, mitigation costs increase due to diminishing returns when measures overlap, with for any measure x:

$$Cost\ new_x = Cost\ old_x{}^* 1/OVcorr_x \qquad (3)$$

Regional differences in mitigation potential are included if these are known. These differences are reflected in the parameters: technical applicability, reduction efficiency, and costs. Partly, these are due to socio-economic circumstances (e.g., different present-day emission intensities and different levels of advancements in farming techniques) that can have short-term implications on mitigation potentials. However, in the case of similar biophysical circumstances across regions, we assume convergence in mitigation potentials (i.e., in minimum emission intensities) in the long term and at maximum carbon prices. Where differences in mitigation potentials are known to be caused by biophysical differences, such as regional temperature, precipitation, geography, etc., this has been taken into account in the form of quantitative constraints of the components underlying the MACs. In this study, we differentiated between regions with high, medium, and low technical applicability for enteric fermentation and $CH_4$ manure (e.g., due to differences in climate and farming systems), based on the GAINS model global $CH_4$ mitigation potentials for livestock in 2030 and 2050[22] (see Supplementary S7). In this assessment, we have estimated the regional technical applicability (TA) on MAC data representing the same measures (with the same RE) across regions, so where a higher reduction potential (RP) was attributable to higher applicability of the measures. Regional differences in reduction efficiency are incorporated in the measure 'anaerobic digestion', which has a higher efficiency in warmer environments. Regional differences in costs are

incorporated where available (see Tables S7.2 and S7.3). It is known that costs can be different across regions, for instance, due to differences in labor costs, costs of capital (with the last two factors typically being negatively correlated), energy and resource requirements and climate-related durability. Unfortunately, in most cases, very little direct information on regional cost differences can be found in literature, in which case we assumed an aggregated global estimate.

The MACs for the agricultural emission sources ($CH_4$ from rice production, $CH_4$ from enteric fermentation in ruminants, $CH_4$ and $N_2O$ from manure, and $N_2O$ from fertilizer) have been constructed fully bottom-up, using the MAC component-based methodology (Eqs. 1 & 2), as was also used in ref. 9. Here, we have updated the agricultural MAC curves by including data on measure-specific reduction efficiency (mainly), technical applicability, cost and source-specific maximum reduction potentials from ±120 studies in combination with the ±80 studies used as a basis for ref. 9. For the Monte Carlo analysis, ranges have been defined for all underlying MAC components, based on the literature review (see Supplementary S7). The newly included studies have been found with a literature search on Scopus, Google Scholar, and Web of Science, using the following keywords: names of emission sources (both agricultural and non-agricultural), names of measures (where known), 'non-$CO_2$', '$CH_4$', '$N_2O$', 'greenhouse gas', 'mitigation', 'reduction', 'measure', 'marginal abatement cost', 'agriculture'. Papers were included if: (1) measures were primarily aimed at emission reduction, (2) results were presented quantitatively and (3) relatable to source-specific MAC components. Most studies, additional to ref. 9, are predominantly published in the 2018–2022 period.

The default MAC curves for the non-agricultural sources are directly based on ref. 9, with only a few, minor modifications to the default values for the maximum reduction potentials (MRPs), where this was justified by the literature review (see Supplementary S6 for a detailed description of the assumptions by source). These central estimates were complemented with optimistic and pessimistic MACs, with MRPs based on the literature study, which were used to scale the default MACs (see Supplementary S5). Waste and industry MACs ($CH_4$ from landfills/solid waste, $CH_4$ from sewage and wastewater, $N_2O$ from adipic and nitric acid production, $N_2O$ from transport, and $N_2O$ from domestic sewage), are based on data up to 2030[24,53–55] but have added assumptions on the technological progress up to 2100, largely based on current best practices[9]. Fossil energy MACs ($CH_4$ from coal, oil and gas production) are based on a dataset from the GAINS model[23,25] with added long-term (MRP) assumptions on including promising technologies that are currently not in use on a large scale. The default F-gas MACs (*hydrofluorocarbons* (HFCs), *perfluorocarbons* (PFCs) and *Sulfur hexafluoride* ($SF_6$)) are directly used from ref. 9, including recent calibrations by refs. 51,56. F-gas emissions and mitigation are endogenously calculated in an IMAGE module, which calculates future F-gas emissions based on economic growth and population data, as well as reductions due to GHG pricing. This study's F-gas calculations are less complex than for the other sources. Mitigation measures are considered complementary (i.e., OVcorr = 100%) and no non-climate policy related reductions are assumed in the baseline (i.e., Bcorr = 0%).

## MAC uncertainty range agricultural sources

The uncertainty analysis for agricultural sources is based on a Monte Carlo (MC) analysis where the underlying parameters have been randomly varied and subsequently run 1000 times. The outcome of the MC analysis is a range in relative reductions at all carbon eq. prices between zero and 4000$/tC. The pessimistic, default and optimistic MACs are based on the 5th, 50th, and 95th percentile in reductions for each carbon price, respectively.

Each MAC component value within a range is given equal weight (i.e., uniform distribution) (see Supplementary S7 for the input values, assumptions, and motivation). The minimum and maximum for the reduction efficiency (RE) component are based on case studies found

**Table 3 | Scenario setup**

| Scenario | NCGG MAC reduction potential | Human GHG-emitting activities | Radiative forcing target 2100 (W/m²) |
|---|---|---|---|
| Base | n.a. | Medium (SSP2) | n.a. |
| 2H | High / Optimistic | Medium (SSP2) | 2.6 |
| 2M | Medium | Medium (SSP2) | 2.6 |
| 2L | Low / Pessimistic | Medium (SSP2) | 2.6 |
| 1.5H | High / Optimistic | Medium (SSP2) | 2.0 |
| 1.5M | Medium | Medium (SSP2) | 2.0 |
| 1.5 L | Low / Pessimistic | Medium (SSP2) | 2.0 |
| 2H_SSP1 | High / Optimistic | Low (SSP1) | 2.6 |
| 2L_SSP3 | Low / Pessimistic | High (SSP3) | 2.6 |

Scenarios are SSP2 based, unless otherwise specified under Scenario. No target is set for Base. The IMAGE SSP2 baseline results in a forcing level of 6.2 W/m² in 2100. 1.5 L and 2L_SSP3 are infeasible scenarios (further discussed in Results).

in the literature. For each measure, the highest and lowest outliers were excluded to prevent the distribution from being skewed. The minimum and maximum of the distributions of the other MAC components are based on a delta value (all in ±% points, since uncertainty is expected to be equally large at high and low values, except for costs, which is given in US\$ and where absolute uncertainty is expected to be proportional to values) around the default component value (unless new information was available, this was based on ref. 9). The default delta values are (in ±%points): TA (40), OVcorr (30), IP (30), TP (10) (note, this applies to the 'diff' term, explained in S7) and (in ±%): Cost(80). The cost delta value is large because of particularly large uncertainty. The values of all components can never be lower than 0 and higher than 100%. Where found relevant, based on existing literature, the sampling was constrained by technical limits (e.g., a TA value is never allowed to be higher than 70% if it is known that 30% of the baseline emissions cannot be reduced by a certain measure).

### MAC uncertainty range non-agricultural sources

The optimistic, default and pessimistic MACs for the non-agricultural sources have been developed by varying the maximum reduction potentials (MRPs) in 2050 and 2100 and scaling them in intermediate years. A full MC analysis is not possible for these sources, since most values of the underlying parameters are unknown, as the short-term MAC data is based on external databases. However, reduction potentials are generally higher, implying lower uncertainty and lower residual emissions in stringent climate scenarios[9,18]. The default MACs are largely equal to those developed by ref. 9, with some small modifications (see Supplementary S5 for the quantitative assumptions by source). Where known, estimates of current technical reduction potentials (based on projections by GAINS and US-EPA[10,22,24]) were used as a minimum value for the pessimistic MACs. This is particularly relevant for F-gases, where emissions, if unmitigated, are estimated to increase to a total of 25% of total NCGG emissions (see Supplementary S3). However, with default assumptions, F-gas emissions are projected to be largely mitigated under stringent climate policy, due to high reduction potentials from well-known technologies[9]. Supplementary S5, therefore, describes possible considerations to lower the F-gas reduction potentials in the pessimistic MAC, to be able to analyze if a substantial increase in residual F-gas emissions in a mitigation scenario could be likely.

### Scenario analysis

The MAC curves have been used as an input to IMAGE 3.2[20,21] in conjunction with Shared Socio-economic Pathway (SSP) based scenario assumptions[43]. The scenarios are described in Table 3. The core set to assess the implications of the MAC uncertainty is based on SSP2, a scenario with middle-of-the-road socio-economic and technological

development assumptions. In these scenarios, a 1.5- and 2-degrees Celsius target should be reached in 2100 (represented by 2.0 W/m² and 2.6 W/m² radiative forcing targets), under optimistic, default and pessimistic NCGG MAC assumptions (i.e., with low (L), medium (M) and high (H) reduction potentials, respectively). The mitigation scenario implications are compared to a no climate policy baseline (Base). Pre-2100 temperature overshoots are allowed. The SSP2-based 2-degree scenarios follow the nationally determined contributions until 2030, followed by fragmented regional climate policy until 2040 and globally concerted climate action until 2100 (i.e., category C3b in the IPCC's scenario classification[44]). The 1.5-degree scenarios are category C2 (allowing a temperature overshoot).

In addition, the analysis includes two additional SSP narratives (in a 2-degree case) to assess the additional uncertainty due to human activities: SSP1 and SSP3, with low and high GHG-emitting activities, respectively. The underlying scenario assumptions for SSP1 and SSP3 are described in ref. 57 with included updates[21]. Next to having lower baseline emissions, the SSP1 mitigation scenarios also include ratcheting up the ambition of the NDCs before 2030, resulting in additional early century emission reductions. SSP1 is combined with optimistic MAC assumptions (H) and SSP3 with pessimistic assumptions (L) to represent the extremes in NCGG emissions. The goal of the scenario analysis is to analyze the effect of MAC uncertainty and uncertainty in human NCGG emitting activities on:

- Feasibility of scenarios
- NCGG emission reductions (total and source-specific)
- Climate policy costs
- Remaining global carbon budgets, i.e., the need for $CO_2$ mitigation

The scenarios used to assess uncertainty in GHG-emitting activities (2H_SSP1 and 2L_SSP3) have been used for the feasibility and carbon budget calculations only. Policy costs and NCGG reduction are not directly comparable due to different cost and baseline emission assumptions.

## Data availability

The optimistic, default and pessimistic $CH_4$ and $N_2O$ MAC curves generated and applied in this study are provided in the Supplementary Data file 1. This data is also directly available in the NAVIGATE database [https://www.navigate-h2020.eu/wp-content/uploads/2022/11/Data_MAC_CH4N2O_Harmsen-et-al_PBL.xlsx].

## Code availability

We provide a stand-alone, Python-based script that can be used to perform the Monte Carlo analysis to build and analyze the agricultural MACs (Supplementary Software 1).

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

## Acknowledgements

This work received funding from the European Union's Horizon 2020 research and innovation program under grant no. 821124 (NAVIGATE).

## Author contributions

M.H., D.v.V., and C.T. designed the study approach. C.T and M.H. performed the literature study. C.T. developed the Monte Carlo Tool. L.H.-I. and P.P. prepared the data from the GAINS model and reviewed the approach and input assumptions for the Monte Carlo analysis and MAC-curve development. C.T. and M.H. developed the MAC curves. C.T. developed Fig. 1. M.H. developed the scenarios with the IMAGE model. M.H., C.T., and F.H. performed the scenario analysis. F.H. developed Fig. 2. M.H. and C.T. wrote the main text. All authors contributed to article review.

## Competing interests

The authors declare no competing interests.
