## [Peer Review File · Nature Communications]

Uncertainty in non-CO₂ greenhouse gas mitigation contributes to ambiguity in global climate policy feasibilityREVIEWER COMMENTS

Reviewer #1 (Remarks to the Author):

This paper developed a set of marginal abatement cost (MAC) curves for non-CO₂ greenhouse gas emissions and evaluated the role of nonCO₂ mitigation in global climate change mitigation scenarios. To quantify the uncertainties in MACs, authors conducted a Monte Carlo analysis for agricultural MACs, a traditionally hard-to-abate sector. Using a well-established integrated assessment model (IMAGE), authors examine non-CO₂ reductions, policy costs, and carbon budget for different 2-degree and 1.5-degree pathways with varying levels of nonCO₂ MACs. This paper suggested that the level of nonCO₂ mitigation could heavily influence of feasibility of mitigation scenarios.

Overall, this is a significant contribution to the field, providing multiple insights into the MAC data, methodology of developing and updating MACs, mitigation scenarios, and policy implications. Because of so many contributions in different aspects that this paper is trying to make, it needs a more detailed and balanced presentation covering all these details. However, the current version focused on presenting details of a few elements (such as agriculture MACs and uncertainties) but lacks information in many other aspects (such as model and scenario assumptions). As a result, this paper looks "unbalanced". Authors could either focus on agriculture MAC as an excellent topical paper or stick to the current scope (all-sector MACs, a few global scenarios). In both cases, it needs substantial revision.

Please see my detailed comments below.

1) The motivation of this paper needs more elaboration. Although it is generally true, some statements might need more timely evidence. For example, In line 43, the authors indicated that "many models (here refers to IAMs) use relatively old information [19, 21]". #19 is a 2015 paper, and #21 is "in press", so I'm unsure whether or not the latest IAMs are still using "relatively old information". To my knowledge, some IAMs have updated their MACs in recent model versions, so authors should be more specific about to what extent IAMs are using old information.

2) The main contribution of this work needs some clarification. From my understanding, this paper improves upon Ref #1 in many aspects but also adapts somewhat similar information from Ref #1. In Methods (second paragraph), authors indicated that "All non-agricultural sources are directly based on [1]". Supplementary S2 is marked as "based on Harmsen et al. 2019".

On the other hand, in Line 56, the authors said, "They have been developed using the method by ref. [1] but complemented with uncertainty ranges and the inclusion of an additional approx. 120 recent studies on mitigation measures", so is this paper directly using the MAC estimates from #1 and additionally developed uncertainty ranges based on ~120 additional papers? Or did this paper use the same method as Ref #1 but create completely new MACs for non-agricultural sources? Also, what are the ~120 recent literatures mean? Although uncommon in this field, a formal systematic review and meta-analysis typically requires clear selection criteria such as literature database, searching keywords, date range, and inclusion/exclusion criteria. An example of such a systematic review is Berrang-Ford et al., 2021 (<https://rdcu.be/c3unL>). While I have no intention to urge authors to follow that rigorous manner, additional clarity in the selection of *recent* evidence (as well as exclusion criteria) considered in this paper will be helpful.

Here I'm trying to determine how much overlap between this paper and Ref #1 because it has been cited numerous times throughout this paper. A Nature Communication paper should be standalone with a sufficiently independent contribution.

3) Method: "Regional cost differences are incorporated based on region-specific cost assessments (Table S5.2 and Table S5.3). But when looking at Table S5.3, most costs are still the same for all regions. So, I hope the authors can explain the regional cost assumptions more. Are they the same

across different regions (even after considering all cost components/determinants), thus leading to the same MACs across regions?

4) Authors argued that “The global 1.5-degree climate target is found to be out of reach under pessimistic MAC assumptions”. This is also implied in the paper title as a “make-or-break for global climate policy”. While nonCO₂ plays an important role, the “make-or-break” and “out of reach” arguments are insufficiently supported by merely one unsolved scenario from one particular model. Indeed, the authors mentioned the carbon budget and reported the carbon budget in Fig.2. However, IPCC AR6’s 1.5-degree has a wider carbon budget range of -90 to 620 Gt (C2: high overshoot, see Table SPM.2 of the IPCC AR6 WGIII SPM). So, if CO₂ were allowed to be further reduced, the 1.5-degree target could be solved by a different model. Similarly, many things could be “make-or-break” for a 1.5-degree future, such as near-term CO₂ reduction rate, level of CDRs, etc. In my perspective, this kind of proactive language might be better reserved for your press release.

5) More details should be provided for scenario assumptions (Method Section 3 Scenario Analysis and Table 1).

First, this paper did not describe any IMAGE background, such as modeling structure, basic solution algorithm, or how policy cost is calculated. All of them might be familiar to IMAGE users, but for non-IMAGE readers, this could be essential. Please move some text from Supplementary S3 into the Method in the main text.

Second, there is no information about CO₂ trajectory and how it was determined. Even for the same carbon budget level, there could be different CO₂ pathways, which might also affect sectoral contributions and fuel switching (fuel switching, in turn, could influence nonCO₂ emissions from fossil production, which are independent of MACs). More specifically, for the two bars in the 1.5-degree column in Figure 2 (both SSP2, 1.5-degree with different MACs), what much of their difference could be attributed to their different nonCO₂ emissions, whereas the remaining difference could be due to their different CO₂ emissions? Please include the time series of your CO₂ and nonCO₂ trajectories in a chart (this paper currently only has two main figures, so space is not a problem here).

Third, there’s little information about peak warming, and many recent non-CO₂ policies and conversations are framed around its role in peak warming (because of methane’s short lifetime). Besides, the peak warming information is also helpful in identifying what type of IPCC 1.5-degree category (C1 or C2) these scenarios are. Line 184-185 has a brief sentence here but chose to “not shown” the peak warming information. Please explain.

6) A main contribution of this paper is the updated agriculture MACs (Fig 1). These estimates for agricultural sectors seem much higher than many other sources (for example, US EPA’s MAC curve, ref #33 in the reference list). Please provide more discussion of why this paper found a much higher agriculture mitigation potential (zero-cost reductions and highest mitigation potentials) than previous literature.

7) There is little discussion about F-gases in both methods and results. The supplementary data also just included MACs for CH₄ and N₂O. Please show more details about this paper’s contribution to F-gas MACs.

Minor points:

1) This paper lacks basic proofreading. Many errors could be easily identified. First, the reference starts in #4 in the Introduction part (line 33), then back to #1 two lines later. So please reorganize the reference order throughout the paper. Second, Online methods part: there are two sub-sections #1s. Third, there are two “Table 1”, one in the main text and another at the end of Method. Forth, in Figure 2 title, 2-degree results are in the left panel, and 1.5-degree are in the right panel. The current text is exactly the opposite. PLEASE do proofreading before submission.

2) Table 1 footnote: “default SSP baseline settings lead to a forcing level of 6.0-6.2 W/m²”. Isn’t this

a deterministic result for a single scenario?

3) The main text suggested that CH₄ rice is a major emitting source, but somehow it was not shown in Figure S1.1.

4) I don't quite follow footnote 2 on Page 9. What does the "lower forcing targets" mean?

5) line 139, "recent studies" needs a reference.

6) SI should have its independent reference list.

7) Table S4.1: US-EPA [30] is dated. Its newer version is #33 in your reference list.

8) Table S9.1: relative reductions based on what baseline.

9) Table S9.2: why 2L has higher reduction than 2M for CH₄ AFOLU?

10) Table S9.2: what's the unit of F-gases

Reviewer #2 (Remarks to the Author):

Comments on Nature Communications Paper "Uncertainty in non-CO₂ greenhouse gas mitigation: Make-or-break for global climate policy feasibility"

Mathijs Harmsen, Charlotte Tabak, Lena Höglund-Isaksson, Florian Humpenöder, Pallav Purohit, Detlef van Vuuren

This paper examines an understudied yet incredibly policy relevant topic of uncertainty in non-CO₂ greenhouse gas mitigation potential. IPCC AR6 WGIII SPM B.1.3 states that, "the current central estimate of the remaining carbon budget from 2020 onwards for limiting warming to 1.5°C with a probability of 50% has been assessed as 500 Gt CO₂," and that the "remaining carbon budgets depend on the amount of non-CO₂ mitigation (± 220 Gt CO₂)." This implies that our current understanding of our ability to mitigate non-CO₂ GHG's leaves a massive uncertainty in just how much CO₂ we can still emit to reach the goals of the Paris Agreement, and thus a huge uncertainty in the cost and feasibility of achieving those goals. Furthermore, the work that is underlying this range assessed by the IPCC is largely based on non-CO₂ MAC curves built up from engineering cost estimates that have a limited ability to represent innovation and technological change in response to high demand. It is common for models to look at technology sensitivities for CO₂ mitigation and explore the role of R&D and policies designed to subsidize technological development, but this type of sensitivity for non-CO₂ GHG abatement technologies is exceedingly rare in the literature. Policy makers are giving greater attention to non-CO₂ gases (e.g. the large number of parties that have signed on to the Global Methane Pledge), and this study will be very influential in future international climate negotiations.

This paper presents a well-designed and methodologically sound approach using a respected model and widely used data set. The information is presented clearly and with sufficient detail to be reproduced and built upon in the literature. The topic is important and timely for international negotiations.

Comments

- Page 10, “the optimistic 1.5-degree scenario is found to have a 0.04-0.05 degree C lower mid-century peak temperature than the default 1.5 case.” This is an important point that could be expanded upon. The vast majority of 1.5C scenarios are peak-and-decline, and even the most stringent C1 scenarios in the AR6 WGIII report allow for up to 0.1C of overshoot before returning to 1.5C. In this context at 0.04-0.05 C reduction in peak temperatures could easily be the difference between a C2 scenario and a C1 scenario. The possibility of deeper NCGG mitigation enabling this kind of ‘peak shaving’ has important implications for policymakers and climate negotiators and would be worth expanding upon here.

- The “Climate policy costs” section on page 10 could be written more clearly. For a given temperature target, moving to the optimistic NCGG scenario results in greater NCGG emissions reductions and creates more head room for CO2 emissions from hard-to-abate sectors. In the language of the next section, the carbon budget for a given target is increased when greater NCGG reductions are possible. This paragraph discusses this by noting that when low-cost options are exhausted climate targets are met by “moving up the MAC curve.” This is somewhat confusing in the context of this paper since the paper is focused on uncertainty in the NCGG MAC curves, but the MAC curve referred to here is the stylized CO2 MAC curve – the need to get more abatement from more costly hard to abate sectors. I’d suggest finding different language to discuss this concept here and build more of a transition to the related discussion of carbon budgets in the next paragraph.

- According to Table S4.1 this paper relies on [US-EPA, United States Environmental Protection Agency (USEPA), Global Mitigation of Non-CO2 Greenhouse Gases: 2010-2030. 2013(EPA-430-R-13-011, Washington DC)] for MAC curves for N2O from adipic acid production, N2O from nitric acid production, CH4 from sewage and wastewater, CH4 from landfills/solid waste. The paper also references [US-EPA, Global Non-CO2 Greenhouse Gas Emission Projections & Mitigation, 2015–2050. 2019, United States Environmental Protection Agency Office of Atmospheric Programs (6207A): Washington] when discussing pessimistic assumptions for non-agricultural sectors. Please clarify which source is used and use the updated EPA (2019) report instead of the outdated EPA (2013) report where appropriate. Also note that by using EPA 2013 the study is selecting more pessimistic MACs as the starting point for waste and industry as that report does not include technological change.

- It appears that for agricultural sources TA is varied across regions based on the relative reduction potential for that region, with regions having lower reductions seeing higher reductions in TA. The introduction states that reduction potentials for non-agriculture sources are generally higher than for agriculture sources, implying lower uncertainty. If this is the rationale for variation of TA, the authors should explain what aspects of TA are being compensated for.

- Additionally, Online Methods, Section 1, Construction of the MAC curves, states that TA can be lower if regional climatic circumstances are unsuitable, and S5 describes differences in RE due to regional climate differences. It would be helpful to clarify what exactly the ranges in TA are adjusting for and how that adjustment is correlated to the relative reduction potential.

Reviewer #1 (Remarks to the Author):

This paper developed a set of marginal abatement cost (MAC) curves for non-CO₂ greenhouse gas emissions and evaluated the role of nonCO₂ mitigation in global climate change mitigation scenarios. To quantify the uncertainties in MACs, authors conducted a Monte Carlo analysis for agricultural MACs, a traditionally hard-to-abate sector. Using a well-established integrated assessment model (IMAGE), authors examine non-CO₂ reductions, policy costs, and carbon budget for different 2-degree and 1.5-degree pathways with varying levels of nonCO₂ MACs. This paper suggested that the level of nonCO₂ mitigation could heavily influence of feasibility of mitigation scenarios.

Overall, this is a significant contribution to the field, providing multiple insights into the MAC data, methodology of developing and updating MACs, mitigation scenarios, and policy implications. Because of so many contributions in different aspects that this paper is trying to make, it needs a more detailed and balanced presentation covering all these details. However, the current version focused on presenting details of a few elements (such as agriculture MACs and uncertainties) but lacks information in many other aspects (such as model and scenario assumptions). As a result, this paper looks “unbalanced”. Authors could either focus on agriculture MAC as an excellent topical paper or stick to the current scope (all-sector MACs, a few global scenarios). In both cases, it needs substantial revision.

- We thank the reviewer for the observation that this work is a significant contribution to the field
- For this study, we feel that the second proposed approach is the most appropriate. We would like to understand how uncertainty in total non-CO₂ GHG mitigation could affect global climate policy feasibility. Therefore, we would like to include a representation of uncertainty in non-agricultural sources as well. The non-agricultural sources are more important in terms of low-cost, high-reduction technical mitigation potentials. For these, we can rely more on studies primarily aimed at technical reduction potentials (US-EPA, GAINS). Although our analysis for these sources has been less elaborate, the aim has been to set plausible minimums and maximums to these sources’ maximum reduction potentials. We have dedicated more extensive analyses to the agricultural sources, as these constitute the highest uncertainty. This more elaborate approach also allows for a full MC analysis, as we have estimates for all underlying MAC parameters.
- We have addressed the unbalance by providing more detailed model and scenario descriptions, as suggested by the reviewer. This mostly concerns an extensive revision of the methods and supplementary information, to describe the approach and assumptions to calculate the non-agricultural MACs, as will be explained in the following comments (note that we have changed the supplement order, based on order of appearance in the text).

Please see my detailed comments below.

1) The motivation of this paper needs more elaboration. Although it is generally true, some statements might need more timely evidence. For example, In line 43, the authors indicated that “many models (here refers to IAMs) use relatively old information [19, 21]”. #19 is a 2015 paper, and #21 is “in press”, so I’m unsure whether or not the latest IAMs are still using “relatively old

information". To my knowledge, some IAMs have updated their MACs in recent model versions, so authors should be more specific about to what extent IAMs are using old information.

- The statement that the MAC curve data is relatively old, mainly followed from the more recent second reference (<https://link.springer.com/article/10.1007/s10584-019-02437-2>), a methane mitigation study (published, not in press, now corrected). Nine leading IAMs took part in that study, in which we made a detailed analysis of the model assumptions on methane mitigation, which included the use of MAC curve datasets (in the paper's SI). In response to the reviewer's comment, for this study, we contacted all participating teams to inquire about their current status of the MAC data used (the references are actually the same for N₂O emissions from the same sources). We found that a few models have updated their MACs (moving to US-EPA, 2019 (e.g. GCAM), or to this study (IMAGE, REMIND-MAGPIE)). However, most of the models still use the datasets as described in the methane mitigation study, which is now almost 4 years old. Roughly two-thirds of the data sources used by the models is 10-20 years old, so the original statement is still found to be valid. We have added the analysis to the supplementary information and added a reference to the SI in the introduction.

2) The main contribution of this work needs some clarification. From my understanding, this paper improves upon Ref #1 in many aspects but also adapts somewhat similar information from Ref #1. In Methods (second paragraph), authors indicated that "All non-agricultural sources are directly based on [1]". Supplementary S2 is marked as "based on Harmsen et al. 2019".

- All agriculture MACs are completely revised. The Monte Carlo analysis is a fundamentally different way of building the MACs, which by itself would already result in a different default MAC curve, even if the same data would have been used (the default MAC in the current approach represents the 50th percentile value in the MC analysis, resulting in a smoother MAC with a somewhat lower reduction potential). However, it now also builds on additional, recent literature (combined with the literature used in the 2019 paper) used to estimate ranges for the MAC components (RE, TA, IP, OVcorr, Costs).
- The MC analysis does rely on the MAC construction method of 2019 paper, but it assumes ranges in the values of the underlying MAC components rather than fixed values.
- The default non-Agriculture MACs are based on the 2019 paper, however, complemented with optimistic and pessimistic MACs, by adjusting the maximum reduction potentials (MRPs) based on literature. Occasionally, the default MRP was also slightly altered if justified by literature.
- We have clarified the text on the non-Agriculture sources in the methods as follows: "The default MAC curves for the non-agricultural sources are directly based on Harmsen et al. (2019) with only a few, minor modifications to the default values for the maximum reduction potentials (MRPs), where this was justified by literature. These central estimates were complemented with optimistic and pessimistic MACs, with MRPs based on the literature study, which were used to scale the default MAC."

On the other hand, in Line 56, the authors said, "They have been developed using the method by ref. [1] but complemented with uncertainty ranges and the inclusion of an additional approx. 120 recent studies on mitigation measures", so is this paper directly using the MAC estimates from #1 and additionally developed uncertainty ranges based on ~120 additional papers? Or did this paper use

the same method as Ref #1 but create completely new MACs for non-agricultural sources? Also, what are the ~120 recent literatures mean? Although uncommon in this field, a formal systematic review and meta-analysis typically requires clear selection criteria such as literature database, searching keywords, date range, and inclusion/exclusion criteria. An example of such a systematic review is Berrang-Ford et al., 2021 (<https://rdcu.be/c3unL>). While I have no intention to urge authors to follow that rigorous manner, additional clarity in the selection of *recent* evidence (as well as exclusion criteria) considered in this paper will be helpful.

Here I'm trying to determine how much overlap between this paper and Ref #1 because it has been cited numerous times throughout this paper. A Nature Communication paper should be standalone with a sufficiently independent contribution.

- This paper is distinctively different than the 2019 paper. A detailed and comprehensive bottom-up approach to estimating an uncertainty range in non-CO₂ MAC mitigation potentials is a completely new approach in this field, as is the scenario analysis based on the "MAC range". In terms of uncertainty analysis, it can be considered a major leap forward beyond existing ex-post, top-down assessments currently used in the IPCC's AR6 and SR15 scenario databases, as well as the single "middle-of-the-road MAC-based" scenario studies. Also new is the application of an MC analysis to build MAC curves, resulting in the completely revised long-term agricultural MACs. The only direct overlap between this study and the 2019 study are the default non-Agriculture MACs.
- The mentioned +/- 120 papers are used in combination with the +/- 80 papers that were used as a basis for the 2019 paper. A large part of the combined total of those papers (so not only the recent 120 papers) are used as input for the MC analysis and are thus used to calculate the uncertainty range for the agricultural sources. "Recent" here implies: studies that predominantly got published after the 2019 paper's finalized analysis; between 2018 and 2022 (although some additional relevant papers were included from before that period, when they were missed in the 2019 paper). To avoid confusion about the use of the additional papers, we have replaced "the most recent literature" in the text by "literature from up to and including 2022" or "a comprehensive literature review", unless we specifically wanted to indicate that we wanted to add more recent papers to the existing dataset.
- 9 papers have been used to help estimate the optimistic and pessimistic 2050 and 2100 maximum reduction potentials (MRPs) for the non-Agricultural sources (See supplement 5, text and Table S5.1) as well as expert insights from the GAINS research group (explained in the introduction, line 70). The main goal of this exercise was to determine a realistic maximum and minimum MRP in 2050 and 2100.
- Note that we added a few references that were not correctly included in the reference list in the previous version of the manuscript (however, they were correctly used in the reduction efficiency calculations). These relate to the measures alternate flooding and drainage to reduce wetland rice CH₄, and Seaweed (*Asparagopsis taxiformis*) to reduce enteric fermentation CH₄.
- We agree that a more formal systematic review / meta-analysis of the literature would have been an improvement. However, we have tried to improve this by being more explicit about which papers have been added and why, following points from the mentioned study as much as possible. We have added the following text to the online methods: "The newly included studies have been found with a literature search on Scopus, Google Scholar, and Web of Science, using the following keywords: names of emission sources (both agricultural and non-agricultural), names of measures (where known), "non-CO₂", CH₄, N₂O, greenhouse gas,

mitigation, reduction, “measure”, “marginal abatement cost”, “agriculture”. Papers were included if: 1) measures were primarily aimed at emission reduction, 2) results were presented quantitatively and 3) relatable to source-specific MAC components. Most studies, additional to ref. [9], are predominantly published in the 2018-2022 period.”

3) Method: “Regional cost differences are incorporated based on region-specific cost assessments (Table S5.2 and Table S5.3). But when looking at Table S5.3, most costs are still the same for all regions. So, I hope the authors can explain the regional cost assumptions more. Are they the same across different regions (even after considering all cost components/determinants), thus leading to the same MACs across regions?

- Unfortunately, cost data is very sparse, and when available, highly uncertain. Only in a minority of the underlying studies that included cost estimates, there was also regional differentiation. Cost estimates, where available, are generally provided on an aggregated global level or are estimated based on a single case study. However, table S5.3 does include some regional differentiation where this was available. That was usually on a higher aggregated region level than the 26 IMAGE world regions. However, note that, even when the cost assumptions are the same for two given regions, the MACs may still strongly differ due to differences in technical applicability or reduction efficiency.
- We have now added the following text to the online methods to better explain the (limitations in) the regional cost assumptions: “Regional differences in costs are incorporated where available (see tables S5.2 and S5.3). It is known that costs can be different across regions, for instance due to differences in labor costs, costs of capital (with the last two factors typically being negatively correlated), energy and resource requirements and climate-related durability. Unfortunately, in most cases, very little direct information on regional cost differences can be found in literature, in which case we assumed an aggregated global estimate.”

4) Authors argued that “The global 1.5-degree climate target is found to be out of reach under pessimistic MAC assumptions”. This is also implied in the paper title as a “make-or-break for global climate policy”. While nonCO2 plays an important role, the “make-or-break” and “out of reach” arguments are insufficiently supported by merely one unsolved scenario from one particular model. Indeed, the authors mentioned the carbon budget and reported the carbon budget in Fig.2. However, IPCC AR6’s 1.5-degree has a wider carbon budget range of -90 to 620 Gt (C2: high overshoot, see Table SPM.2 of the IPCC AR6 WGIII SPM). So, if CO2 were allowed to be further reduced, the 1.5-degree target could be solved by a different model. Similarly, many things could be “make-or-break” for a 1.5-degree future, such as near-term CO2 reduction rate, level of CDRs, etc. In my perspective, this kind of proactive language might be better reserved for your press release.

- We agree with the reviewer that this should be stated differently. We have now indicated in the results that these conclusions are based on the IMAGE model setup. We have mentioned that model comparisons have shown that compared to other models, IMAGE can be regarded as “average” in terms of inertia/speed of implementation and energy system transformation (we now refer to a recent diagnostic study). This (as well as the mentioned carbon budget range) automatically implies that some models may still find the 1.5 degree target within reach – based on more optimistic assumptions. Moreover, also other mitigation strategies

currently not explored frequently by IAMs (such as circular economy) could make the target feasible.

- We also agree that, given that the world is close to exceeding 1.5 degrees warming, there are multiple factors that can be considered “make-or-break” for reaching the 1.5-degree target. We have added a comment to the results to stress this, indicating that for instance (as indeed mentioned by the reviewer) near-term CO₂ reduction rate and the level of CDR are crucial factors here as well.
- However, we do think the paper title is justified. The feasibility of global climate policy goes beyond reaching the 1.5-degree target only. The uncertainty range (and therefore the projected range in projected policy costs and CO₂ emissions, under a given stringent target) is very large. That would be a universal outcome across models.

5) More details should be provided for scenario assumptions (Method Section 3 Scenario Analysis and Table 1).

First, this paper did not describe any IMAGE background, such as modeling structure, basic solution algorithm, or how policy cost is calculated. All of them might be familiar to IMAGE users, but for non-IMAGE readers, this could be essential. Please move some text from Supplementary S3 into the Method in the main text.

Second, there is no information about CO₂ trajectory and how it was determined. Even for the same carbon budget level, there could be different CO₂ pathways, which might also affect sectoral contributions and fuel switching (fuel switching, in turn, could influence nonCO₂ emissions from fossil production, which are independent of MACs). More specifically, for the two bars in the 1.5-degree column in Figure 2 (both SSP2, 1.5-degree with different MACs), what much of their difference could be attributed to their different nonCO₂ emissions, whereas the remaining difference could be due to their different CO₂ emissions? Please include the time series of your CO₂ and nonCO₂ trajectories in a chart (this paper currently only has two main figures, so space is not a problem here).

Third, there's little information about peak warming, and many recent non-CO₂ policies and conversations are framed around its role in peak warming (because of methane's short lifetime). Besides, the peak warming information is also helpful in identifying what type of IPCC 1.5-degree category (C1 or C2) these scenarios are. Line 184-185 has a brief sentence here but chose to “not shown” the peak warming information. Please explain.

- We have now added an elaborate model description to the online methods (in 1 System boundaries), explaining the main model structure (ecological-environmental IAM, partial equilibrium, simulation model/no foresight, spatial resolution), the solution algorithm used in the climate policy module and the policy cost calculation (with policy costs representing first-order expenditures).
- On the second point: We have added the CO₂ and non-CO₂ emission trajectories as well as the radiative forcing and global mean temperature profiles to the supplementary information (S10) (and referred to it in the main text). We found the SI a more suitable place for these results, as we have not found any outcomes that are crucial to support the main storyline, where we would like to focus on non-CO₂ and the main feasibility indicators. Space might also be an issue, now that we have added quite some additional information to the main text.
- With the emission trajectories, we show that the pathways and mitigation strategies are relatively similar across the scenarios (given a specific climate target and SSP baseline). CO₂ directly compensates for non-CO₂ in a similar way in the different scenarios (the exception being the SSP1-based 2-degree scenario, which is characterized by lower emissions in general

and earlier-century emission reductions, due to earlier allowed climate action (ratcheting up the NDCs) and lower baseline emissions.

- On the third point, we have added more detailed info on peak warming and forcing to the SI and refer to this in the results. We found that the earlier conclusion, suggesting that the optimistic MACs led to earlier action in a 1.5-degree case, resulting in a lower peak temperature, was incorrect. The scenario setup for 1.5M was different than for 1.5H (in the last version, 1.5M was forced towards delayed climate action). We have corrected that with a rerun of 1.5M. With a consistent scenario setup, we did not find a significant difference in peak temperature (and forcing) between the scenarios. The reason for this (which we now also explain in the results) is that the MACs represent a relatively low, short-term reduction potential, leading to relatively small differences between MACs in the short term (until 2030-2040, when maximum forcing is reached). The 2-degree scenarios also follow the NDCs up till 2030, with fragmented policy until 2040, further limiting variation in early-century strategies. However, we do show that peak temperature is found to be slightly (0.02 degrees) lower in the 2H_SSP1 case, due to the beforementioned differences in baseline and NDC assumptions. We now also note in the Results that the impact of NCGG mitigation potential on peak temperature is model- and scenario dependent and could thus be further explored in a multi-model study.
- On the IPCC scenario classification: we have now provided the categories in the introduction and method; C3b (=NDCs + 2-degrees) for the 2-degrees scenarios and C2 (1.5 degrees with overshoot) for the 1.5 degrees scenarios. In the results, we now show that the carbon budgets are in line with the carbon budget ranges for these categories, as shown in AR6.

6) A main contribution of this paper is the updated agriculture MACs (Fig 1). These estimates for agricultural sectors seem much higher than many other sources (for example, US EPA's MAC curve, ref #33 in the reference list). Please provide more discussion of why this paper found a much higher agriculture mitigation potential (zero-cost reductions and highest mitigation potentials) than previous literature.

- We agree that it is important that we explain the critical differences between the NCGG MAC curves in this study and those developed by US-EPA and GAINS. The latter MAC datasets mainly represent the present-day technical reduction potentials as measured in multiple case studies, although they do account for modest technical progress towards 2050, yet not for changes in the level of technology acceptance.
- This and the following explanation of the difference between the MAC studies is now added to the discussion.
- In this study, we deliberately fully account for all future technological change and removal of non-technical implementation barriers under stringent climate policy conditions. The longer-term (up to 2100) perspective of this study also requires that these factors are included, including the high uncertainty that comes with them. As these factors contribute to more effective mitigation, this study's default MAC curves generally represent higher reduction potentials, while this study's pessimistic MACs are generally found to be in line with US-EPA and GAINS (when looking at 2050, the most future year where the MACs can be compared). This fits well with the assumption that present day reduction potentials should at the very least be reachable in any future scenario, as with the prerequisite that this study's MAC range should span the full potential solution space.

- As an example of the last statement; the US-EPA 2019 and GAINS (Höglund-Issakson, 2020) estimates for the reduction potential for enteric fermentation in 2050 are 9% and 14-15%, respectively. In this study's pessimistic MAC it is 11%.

7) There is little discussion about F-gases in both methods and results. The supplementary data also just included MACs for CH₄ and N₂O. Please show more details about this paper's contribution to F-gas MACs.

- We have added several descriptions to the main text and supplement to explain this better. Most importantly, in supplement S6 (Emission source-specific measures and assumptions) we have added a description of the literature and assumptions underlying the default MAC curves, and in supplement S5 (Input parameters pessimistic – default – optimistic MACs – non-Agriculture) we have extensively described the argumentation behind the minimum MRP values in the pessimistic MACs. These are considered the most relevant, since a revised, lower reduction potential could lead to substantially higher emissions (unmitigated F-gas emissions, in a no policy baseline, form a large share of total non-CO₂ emissions (25% in the IMAGE SSP2 in 2100)). In the methods, we have added a more detailed description of the F-gas module in IMAGE as well as an explanation of the relevance of knowing the lower side of the mitigation potential range and assessing its implications. In the results, we now explain in more detail that, even with pessimistic assumptions, residual F-gas emissions are found to be low. Uncertainty is somewhat higher for PFCs and SF₆, but the implications are modest since these gases only comprise a small part of the total emissions.

Minor points:

1) This paper lacks basic proofreading. Many errors could be easily identified. First, the reference starts in #4 in the Introduction part (line 33), then back to #1 two lines later. So please reorganize the reference order throughout the paper. Second, Online methods part: there are two sub-sections #1s. Third, there are two "Table 1", one in the main text and another at the end of Method. Forth, in Figure 2 title, 2-degree results are in the left panel, and 1.5-degree are in the right panel. The current text is exactly the opposite. PLEASE do proofreading before submission.

- We would like to thank the reviewer for finding these errors. They have all been corrected. Regarding the numbering of the tables. We were not sure if the numbering continues in the online methods. We now followed the suggestion of the reviewer.

2) Table 1 footnote: "default SSP baseline settings lead to a forcing level of 6.0-6.2 W/m²". Isn't this a deterministic result for a single scenario?

- There is indeed a single value for the SSP2 baseline (6.2 W/m² in 2100). The comment was added to indicate that this is in line with radiative forcing values projected by other models. After consideration, we think it's most relevant to indicate the value in this study, so the note is now changed to: "The IMAGE SSP2 baseline results in a forcing of 6.2 W/m²"

3) The main text suggested that CH₄ rice is a major emitting source, but somehow it was not shown in Figure S1.1.

- It is in fact shown in the figure, but as “wetland rice”. We have changed the name in the accompanying table for consistency.

4) I don't quite follow footnote 2 on Page 9. What does the “lower forcing targets” mean?

- This is now corrected, following the suggestion of the reviewer by using the IPCC scenario classification. We shown that the carbon budgets actually fit well within the carbon budget ranges of the scenario categories, as provided in AR6. However, we do explain that the 2.0 W/m² forcing target in our 1.5-degree scenario corresponds with a >67% chance of staying below 1.5-degrees, whereas the C2 scenario category also allows for scenarios aimed at limiting warming by >50% .

5) line 139, “recent studies” needs a reference.

- References have now been added.

6) SI should have its independent reference list.

- This has been corrected. Both the main text and the supplement have a separate reference list.

7) Table S4.1: US-EPA [30] is dated. Its newer version is #33 in your reference list.

- This is now corrected. We should have referred to the newer version.

8) Table S9.1: relative reductions based on what baseline.

- Compared to SSP2, this is now added

9) Table S9.2: why 2L has higher reduction than 2M for CH₄ AFOLU?

- This was indeed an error. We thank the reviewer for finding it. The emissions are indeed lower in 2L. We also checked the values in the rest of the table and found no further inconsistencies.

10) Table S9.2: what's the unit of F-gases

- The unit is Gt CO2 eq. It is now correctly mentioned.

Reviewer #2 (Remarks to the Author):

Comments on Nature Communications Paper “Uncertainty in non-CO2 greenhouse gas mitigation: Make-or-break for global climate policy feasibility”

Mathijs Harmsen, Charlotte Tabak, Lena Höglund-Isaksson, Florian Humpenöder, Pallav Purohit, Detlef van Vuuren

This paper examines an understudied yet incredibly policy relevant topic of uncertainty in non-CO2 greenhouse gas mitigation potential. IPCC AR6 WGIII SPM B.1.3 states that, “the current central estimate of the remaining carbon budget from 2020 onwards for limiting warming to 1.5°C with a probability of 50% has been assessed as 500 Gt CO₂,” and that the “remaining carbon budgets depend on the amount of non-CO₂ mitigation (± 220 Gt CO₂).” This implies that our current understanding of our ability to mitigate non-CO₂ GHG’s leaves a massive uncertainty in just how much CO₂ we can still emit to reach the goals of the Paris Agreement, and thus a huge uncertainty in the cost and feasibility of achieving those goals. Furthermore, the work that is underlying this range assessed by the IPCC is largely based on non-CO₂ MAC curves built up from engineering cost estimates that have a limited ability to represent innovation and technological change in response to high demand. It is common for models to look at technology sensitivities for CO₂ mitigation and explore the role of R&D and policies designed to subsidize technological development, but this type of sensitivity for non-CO₂ GHG abatement technologies is exceedingly rare in the literature. Policy makers are giving greater attention to non-CO₂ gases (e.g. the large number of parties that have signed on to the Global Methane Pledge), and this study will be very influential in future international climate negotiations.

- We want to thank the reviewer for stressing the high importance and potential of this study.

This paper presents a well-designed and methodologically sound approach using a respected model and widely used data set. The information is presented clearly and with sufficient detail to be reproduced and built upon in the literature. The topic is important and timely for international negotiations.

- We thank the reviewer for underlining the paper’s methodological quality, writing, literature use and relevance for future climate negotiations.

Comments

- Page 10, “the optimistic 1.5-degree scenario is found to have a 0.04-0.05 degree C lower mid-century peak temperature than the default 1.5 case.” This is an important point that could be expanded upon. The vast majority of 1.5C scenarios are peak-and-decline, and even the most stringent C1 scenarios in the AR6 WGIII report allow for up to 0.1C of overshoot before returning to 1.5C. In this context at 0.04-0.05 C reduction in peak temperatures could easily be the difference between a C2 scenario and a C1 scenario. The possibility of deeper NCGG mitigation enabling this kind of ‘peak shaving’ has important implications for policymakers and climate negotiators and would be worth expanding upon here.

- It is very a good suggestion to dive deeper in the peak temperature differences. We have done this by explaining scenario differences in more detail in the results, based on the CO₂ and non-CO₂ emission trajectories, radiative forcing and global mean temperature profiles, which we now added to the supplementary information (S10). However, we needed to revise our earlier (tentative) outcome that the optimistic MAC led to a lower peak temperature. As mentioned in the response to reviewer 1, we found an inconsistency in the scenario setup for 1.5M and reran this scenario with the same setup as for 1.5H (in the last version, we inadvertently forced 1.5M into delayed climate action). With the rerun, we did not find significant scenario differences for peak temperature and peak radiative forcing (both across the SSP2-based 2-degree and 1.5-degree scenarios). The reason for this (which we now also explain in the results) is that the MACs represent a relatively low, short-term reduction potential, leading to relatively small differences between MACs in the short term (until 2030-2040, when maximum forcing is reached). The 2-degree scenarios also follow the NDCs up till 2030, with fragmented policy until 2040, further limiting variation in early-century strategies. However, we do show that peak temperature is found to be slightly (0.02 degrees) lower in the 2H_SSP1 case, due to the beforementioned differences in baseline and NDC assumptions. We further note in the results that the impact of NCGG mitigation potential on peak temperature is model- and scenario dependent and could thus be further explored in a multi-model study.

- The “Climate policy costs” section on page 10 could be written more clearly. For a given temperature target, moving to the optimistic NCGG scenario results in greater NCGG emissions reductions and creates more head room for CO₂ emissions from hard-to-abate sectors. In the language of the next section, the carbon budget for a given target is increased when greater NCGG reductions are possible. This paragraph discusses this by noting that when low-cost options are exhausted climate targets are met by “moving up the MAC curve.” This is somewhat confusing in the context of this paper since the paper is focused on uncertainty in the NCGG MAC curves, but the MAC curve referred to here is the stylized CO₂ MAC curve – the need to get more abatement from more costly hard to abate sectors. I’d suggest finding different language to discuss this concept here and build more of a transition to the related discussion of carbon budgets in the next paragraph.

- We have rephrased this as follows: “Global climate policy costs (Figure 2, panel b) strongly depend on the availability of NCGG mitigation options, which are on average lower in cost than CO₂ mitigation options [9], but also expand the range of possible measures. When low-cost options are exhausted sooner (i.e., in the pessimistic MAC case), climate targets can only be met by applying higher-cost mitigation measures (both for CO₂ and NCGG emissions).”
- So we have removed “moving up the MAC curve”, which could indeed be confusing, since it does not directly refer to the MAC curves developed in this study. However, note that it does not only refer to CO₂. Also more expensive NCGG emissions need to be reduced when options are exhausted (now also mentioned in the description).

- According to Table S4.1 this paper relies on [US-EPA, United States Environmental Protection Agency (USEPA), Global Mitigation of Non-CO₂ Greenhouse Gases: 2010-2030. 2013(EPA-430-R-13-011, Washington DC)] for MAC curves for N₂O from adipic acid production, N₂O from nitric acid production, CH₄ from sewage and wastewater, CH₄ from landfills/solid waste. The paper also references [US-EPA, Global Non-CO₂ Greenhouse Gas

Emission Projections & Mitigation, 2015–2050. 2019, United States Environmental Protection Agency Office of Atmospheric Programs (6207A): Washington] when discussing pessimistic assumptions for non-agricultural sectors. Please clarify which source is used and use the updated EPA (2019) report instead of the outdated EPA (2013) report where appropriate. Also note that by using EPA 2013 the study is selecting more pessimistic MACs as the starting point for waste and industry as that report does not include technological change.

- This was an incorrect reference. US-EPA (2019) has been used in the latest version of the model for the mentioned sources. This has now been corrected throughout the text.
-
- It appears that for agricultural sources TA is varied across regions based on the relative reduction potential for that region, with regions having lower reductions seeing higher reductions in TA. The introduction states that reduction potentials for non-agriculture sources are generally higher than for agriculture sources, implying lower uncertainty. If this is the rationale for variation of TA, the authors should explain what aspects of TA are being compensated for.
 - We have tried to roughly derive regional differences in TA from GAINS. We took the GAINS MAC data that represented regional reduction potentials (RPs) when using the same measures (so measures with the same reduction efficiencies (RE) when a measure is applied). This meant that if the projected RP for a region was lower, it was the result of lower application of the measure in a certain region. So a lower reduction potential (RP) in a specific region corresponds with a lower TA, and a higher RP with a higher TA. In the reviewer's comment it seems to be suggested that this might be the other way around. We have therefore clarified this now in the Methods. We further refer to the supplement where we explain the approach in more detail. In both the Methods and SI, we now explain that the variation in TA (as found in the GAINS dataset) is primarily the result of differences in climate and farming systems. Note that the assumptions on TA are highly uncertain and just a rough estimate. The MC analysis and resulting MAC range is therefore crucial to capture that uncertainty.
 - In the introduction, we have now also mentioned that measures for non-agricultural sources generally have higher RE and TA values (i.e., have more reduction when applied and are more applicable (i.e., they target a larger share of the source emissions)).
 - Additionally, Online Methods, Section 1, Construction of the MAC curves, states that TA can be lower if regional climatic circumstances are unsuitable, and S5 describes differences in RE due to regional climate differences. It would be helpful to clarify what exactly the ranges in TA are adjusting for and how that adjustment is correlated to the relative reduction potential.
 - The range in TA should represent the uncertainty in the level of technical applicability (next to due to climatic factors, also due to differences in farming systems, which we now also mention in the methods). The range in RE should represent the uncertainty in emission reductions when a measure can be applied. Note that both TA and RE can be regionally different due to climatic circumstances. An example of a climatic factor lowering the TA value is how a warmer climate excludes the use of some types of low-methane producing cattle (e.g. Holstein cows). An example of a climatic factor affecting the RE is how a warmer climate

contributes to the efficiency of anaerobic digesters to reduce CH₄ emissions from manure. So a both a higher RE as well as a higher TA correlates with a higher reduction potential. We have now clarified this in the methods.

REVIEWERS' COMMENTS:

Reviewer #1 (Remarks to the Author):

I appreciate authors' effort to address my comments, and all clarifications/revisions make sense to me. I have no further questions.

Reviewer #2 (Remarks to the Author):

I would like to thank the authors for thoughtfully addressing all comments from my review. This paper makes an important contribution to a policy relevant topic and I have no reservations about publication of this work.

REVIEWERS' COMMENTS:

Reviewer #1 (Remarks to the Author):

I appreciate authors' effort to address my comments, and all clarifications/revisions make sense to me. I have no further questions.

- We would like to thank the reviewer for the very constructive and useful comments

Reviewer #2 (Remarks to the Author):

I would like to thank the authors for thoughtfully addressing all comments from my review. This paper makes an important contribution to a policy relevant topic and I have no reservations about publication of this work.

- We would like to thank the reviewer for the very constructive and useful comments